# The Effects of Whole-Body Vibration on Spasticity in Stroke: A Systematic Review and Meta-Analysis

**DOI:** 10.3390/jcm14175966

**Published:** 2025-08-23

**Authors:** Jeong-Woo Seo, Jung-Dae Kim, Ji-Woo Seok

**Affiliations:** Digital Health Research Division, Korea Institute of Oriental Medicine, 1672 Yuseong-daero, Yuseong-gu, Daejeon 34054, Republic of Korea; jwseo02@kiom.re.kr

**Keywords:** stroke, spasticity, vibration therapy, whole-body vibration, meta-analysis

## Abstract

**Background/Objectives**: Spasticity is a common and disabling sequela of stroke that limits voluntary movement and functional recovery. Vibration therapy (VT) has been proposed as a non-invasive neuromodulatory intervention, but the existing studies report inconsistent outcomes due to methodological heterogeneity. This study aimed to evaluate the overall effectiveness of VT in reducing post-stroke spasticity and to identify optimal stimulation parameters via meta-analytic and meta-regression approaches. **Methods**: A systematic review and meta-analysis were conducted following the PRISMA 2020 guidelines. Standardized effect sizes (Hedges’ g) were calculated based on the within-group pre–post changes and compared across the groups. Meta-regression and subgroup analyses explored seven potential moderators, including the vibration frequency, amplitude, and time since stroke onset. **Results**: Thirteen randomized controlled trials (RCTs) involving whole-body or focal vibration interventions in stroke populations were included. Vibration therapy significantly reduced spasticity, yielding a moderate overall effect size (Hedges’ g = −0.50; 95% CI: −0.65 to −0.34; *p* < 0.001). The greatest treatment effects were observed when VT was applied during the late subacute to early chronic phase (6–12 months post-stroke), with low-frequency (<20 Hz) and low-amplitude (≤0.5 mm) stimulation. The frequency, amplitude, and stroke onset emerged as significant moderators (*p* < 0.05). **Conclusions**: Vibration therapy is an effective and clinically meaningful intervention for post-stroke spasticity, particularly when delivered with low-intensity parameters during the optimal recovery window. These findings support the development of individualized VT protocols and provide evidence to guide future rehabilitation strategies.

## 1. Introduction

Spasticity is a frequent and debilitating sequela of stroke, characterized by a velocity-dependent increase in muscle tone due to hyperexcitability of the stretch reflex. This pathophysiological condition impairs voluntary movement, restricts joint mobility, and hinders functional recovery, thereby markedly diminishing patients’ quality of life [1].

Despite the availability of various treatment modalities—including pharmacologic agents, physical therapy, and surgical interventions—these approaches are often limited by short-lived efficacy, adverse effects, and financial or logistical constraints [2]. For instance, oral antispastic agents such as baclofen and tizanidine are frequently associated with adverse effects including drowsiness, hypotension, and generalized muscle weakness, which restrict their long-term use [3]. Botulinum toxin injections, although effective in the short term, carry the risk of neutralizing antibody formation and impose a considerable cost [4]. Surgical options such as selective neurotomy or tenotomy, while effective in selected cases, are invasive, require prolonged recovery, and carry a risk of postoperative complications [5]. These limitations underscore the need for safer, non-invasive, and sustainable alternatives for the management of post-stroke spasticity.

In response, several non-invasive modalities with favorable safety profiles and potential for repetitive application have gained prominence. Among these are functional electrical stimulation (FES), transcranial magnetic stimulation (TMS), and transcutaneous electrical nerve stimulation (TENS) [6]. FES promotes muscle re-education and neuroplasticity by inducing muscle contraction through peripheral nerve stimulation, but it may induce discomfort, skin irritation, or muscle fatigue, thereby limiting the patient adherence. TMS, which modulates cortical excitability through magnetic induction, has shown promise in inducing neural reorganization; however, its clinical utility is restricted by the need for expensive equipment and trained personnel and the risk of side effects such as headache or, in rare cases, seizure. TENS, which applies low-intensity electrical currents via surface electrodes, is used to manage pain and mild spasticity, but its therapeutic efficacy remains inconsistent across individuals, with a tendency for decreased effectiveness over time. Each of these modalities offers distinct mechanisms of action and therapeutic benefits, yet they also present limitations that may hinder their widespread clinical use.

Recently, vibration therapy (VT) has emerged as a promising alternative that addresses many of these limitations. Vibration interventions—delivered via either whole-body vibration (WBV) or focal muscle vibration (FMV)—apply mechanical oscillations to muscles or joints, thereby providing somatosensory input to the central and peripheral nervous systems. This input is thought to reduce spasticity by modulating the spinal reflex circuitry, enhancing presynaptic inhibition, and altering cortical excitability [7]. VT is non-invasive, largely free of discomfort, and operable with relatively simple devices, offering high feasibility and accessibility in clinical environments. Unlike FES, it does not elicit direct muscle contractions, reducing treatment-related fatigue and discomfort. Compared to TMS, it is less resource-intensive and more amenable to routine application. These attributes render VT particularly suitable for older adults and individuals with severe impairment who are unable to participate in conventional exercise-based rehabilitation.

A number of clinical studies have investigated the effects of VT on post-stroke spasticity. For example, one study reported that FMV reduced upper limb spasticity and facilitated motor recovery in patients with chronic stroke [8], while another study demonstrated partial benefits of WBV for lower limb spasticity [9]. However, the findings across studies remain heterogeneous due to discrepancies in their participant characteristics (e.g., chronic vs. subacute stroke), vibration parameters (e.g., frequency, amplitude, posture), and outcome measures. Although recent meta-analyses have attempted to synthesize the evidence on VT in stroke populations, important methodological limitations persist.

Regarding the key distinctions between this study and previous meta-analyses, the first lies in the analytic strategy employed. Most prior meta-analyses assessed intervention effects by simply comparing the post-intervention group means—an approach that fails to control for the baseline variability, such as differences in the initial scores arising from heterogeneity in the participant characteristics, stroke severity, or pre-intervention spasticity levels—thereby limiting the precision and validity of treatment effect estimates [9]. To address this limitation, the present study calculated effect sizes based on the within-group pre-to-post change scores, which captured the actual improvement attributable to the intervention, and then compared these change scores between groups using a pretest–posttest control group design. This methodology adjusted for the baseline heterogeneity, increased the statistical rigor, and yielded more accurate and reliable estimates of the net intervention effect.

The second distinction lies in the scope of the research. Previous meta-analyses often pooled diverse outcome measures—including clinical scales, physiological indices, and functional performance measures—which inflated statistical heterogeneity and reduced the clinical interpretability of the pooled results [10]. In contrast, the current study deliberately restricted the outcomes to scores on the Modified Ashworth Scale (MAS) and Modified Tardieu Scale (MTS), the most widely accepted and validated clinical tools for assessing spasticity in rehabilitation. This focused approach improved the data homogeneity, reduced the unexplained variance, and enhanced the clinical applicability of the findings. Additionally, the analysis incorporated a comprehensive set of seven moderator variables—the amplitude, frequency, application posture, stroke stage, total dosage, vibration type (WBV vs. FMV), and outcome type (MAS vs. MTS)—providing a more nuanced examination than previous reviews.

A third important distinction is the emphasis on the clinical impact. Rather than merely reporting the average treatment effects, this study identified optimal vibration parameters and the most effective application window for maximizing the therapeutic benefit through meta-regression and subgroup analyses. These findings offer actionable guidance for clinicians in designing individualized, evidence-based vibration therapy protocols tailored to specific patient profiles.

Taken together, this study provides a methodologically robust and clinically relevant synthesis of the effects of VT on post-stroke spasticity. By controlling for baseline variation, minimizing the heterogeneity in the outcome measures, and identifying key moderating variables, it delivers a strong evidence base to inform the optimization and personalization of vibration-based rehabilitation protocols for stroke survivors with spasticity.

## 2. Methods

### 2.1. Study Design

This research presents a systematic review and meta-analysis aimed at comprehensively evaluating and statistically synthesizing the effects of VT in reducing spasticity following a stroke. This research was pre-registered in PROSPERO (registration number: CRD420251067889) and was conducted in accordance with the Preferred Reporting Items for Systematic Reviews and Meta-Analyses (PRISMA) 2020 reporting guidelines [10,11]. The aim of this research was to quantitatively assess the impact of VT on post-stroke spasticity. To this purpose, we considered research designs that allowed for both within- and between-group pre- and post-intervention comparisons, such as randomized controlled trials (RCTs), non-randomized controlled trials (nRCTs), and crossover studies, provided that they included appropriate control groups. We ruled out single-arm (pre–post) designs, case reports, and non-comparative observational studies.

### 2.2. Literature Search and Screening

The literature search was conducted in five major databases: the Cochrane Central Register of Controlled Trials (CENTRAL), Embase (Ovid), ScienceDirect, PubMed, and Web of Science. There were no restrictions on the search language or search period, and the literature published up to June 2025 was included. Unpublished studies were not included. The search language was limited to English.

The search strategy was structured as follows based on the PICO elements (population, intervention, comparison, outcome): (1) the population (stroke OR cerebrovascular accident OR post-stroke OR hemiplegia OR hemiparesis), (2) the intervention (vibration therapy OR whole body vibration OR focal muscle vibration OR mechanical vibration), (3) the comparison group (C) (sham OR no treatment OR usual care OR placebo OR treatment as usual OR TAU OR waitlist), and the outcome (O) (spasticity OR Modified Ashworth Scale OR MAS OR Modified Tardieu Scale OR MTS OR stiffness). The terms used were combined using Boolean operators (e.g., AND, OR), and searches were conducted in the title, abstract, and keyword fields. To identify additional eligible studies, backward and forward citation tracking, reference list screening, and expert consultation were employed.

The inclusion of studies was independently assessed by two authors (J.-W.Seo & J.-D.K.), and disagreements were resolved through discussion or consultation with a third reviewer. For studies meeting the inclusion criteria, the relevant data were extracted from the published reports, and in cases of insufficient or missing data, the original authors were contacted to request further information.

### 2.3. Eligibility Criteria

The selection of the literature was performed according to the predefined inclusion and exclusion criteria, which were categorized into five domains: the subject characteristics, type of intervention, comparison group, outcome indicators, and study design.

First, the subjects (population) included adults diagnosed with ischemic or hemorrhagic stroke and were limited to cases in which spasticity was clinically confirmed in the upper or lower limbs. There were no restrictions on the duration of the stroke, such as acute, subacute, or chronic. On the other hand, individuals with spasticity of a non-stroke origin (e.g., cerebral palsy, traumatic brain injury, multiple sclerosis), participants under 18 years of age, and animal or in vitro studies were excluded.

The intervention was defined as mechanically applied VT, which included whole-body vibration and focal muscle vibration. The included studies needed to clearly report key parameters of the intervention, such as the frequency, amplitude, and application time of vibration, and all interventions applied alone or as an adjunct to a standard rehabilitation program were included. On the other hand, interventions that included non-mechanical modalities such as ultrasound, electromagnetic waves, or auditory stimulation, or interventions that did not include vibration as a main component or did not directly measure spasticity were excluded.

The comparison groups underwent placebo vibration, no treatment, or general rehabilitation, were on a waiting list, or were active control groups (e.g., stretching, task-oriented training, etc.), and crossover design studies meeting these conditions were also included. On the other hand, studies with poorly defined comparison groups or studies in which VT and other interventions were applied in combination, making it impossible to independently confirm the effect of vibration, were excluded.

The outcome indicators were limited to values that quantitatively evaluated the degree of spasticity and mainly included the Modified Ashworth Scale (MAS) and Modified Tardieu Scale (MTS). However, studies that reported only functional indicators unrelated to spasticity were excluded from the analysis.

Finally, the study design included comparative studies that allowed for both pre- and post-intervention comparisons and between-group comparisons, such as randomized controlled trials (RCTs), non-randomized controlled trials (nRCTs), and crossover trials. Single-group pre- and post-intervention comparison designs, case reports and descriptive studies, observational studies without control groups, and studies that did not allow for quantitative analysis (e.g., due to missing means and standard deviations) were excluded. A list of excluded studies and reasons for exclusion and a PRISMA checklist are provided as “Appendix A”.

### 2.4. Data Extraction

Data extraction was performed in a structured manner according to predefined criteria. The extracted information included the study design (e.g., randomized controlled trial, crossover design, etc.), number of participants and mean participant age, clinical characteristics (e.g., diagnosis, affected body region), vibration intervention conditions (position during vibration application, duration of one intervention session, number of intervention sessions per week, total intervention period), vibration settings (frequency, amplitude, gravitational acceleration, etc.), intervention type for the control group (e.g., no intervention, sham vibration, general rehabilitation exercise, etc.), and spasticity assessment tools and measured outcomes (e.g., MAS, MTS, muscle tone score, etc.).

In particular, key variables related to the vibration intervention were recorded according to the ‘Body Vibration Big Five’ criteria proposed by Oroszi et al. [7], which include the following five key elements: (1) the vibration frequency (Hz), (2) amplitude (mm), (3) application method (type of platform or posture), (4) duration of one intervention session and number of intervention sessions per week, and (5) total intervention period.

When the research results were presented in the form of images (e.g., graphs) rather than tables, the data were extracted using WebPlotDigitizer software (ver.5, Automeris LLC, USA). In addition, when statistical values such as the mean or standard deviation were not clearly reported, additional information was requested from the original author via email. Data extraction was performed independently by two researchers, and in cases of disagreement over the interpretation or figures, a third reviewer resolved these discrepancies through consensus.

### 2.5. Assessment of Methodological Quality

The methodological quality of each included study was assessed using the standard tool Risk of Bias 2 (RoB 2) developed by Cochrane [12,13]. Two versions of the assessment tool were used depending on the study design. The general RoB 2 tool was applied to randomized controlled trials (RCTs) and non-randomized controlled trials (nRCTs), and the extended version of RoB 2 for crossover trials was applied to crossover studies.

The RoB 2 tool assesses the risk of bias in the following five domains: (1) bias in the randomization process, (2) bias due to deviations from the intended interventions, (3) bias due to missing outcome data, (4) bias in measurement of the outcome, and (5) bias due to selective reporting. Meanwhile, in the case of crossover studies, in addition to the above items, an additional domain (Domain S) related to the period and carryover effects was also assessed. This domain is important for addressing the potential carryover between periods, which must be considered due to the inherent structure of crossover designs.

Each domain was rated as having a ‘low risk of bias’, ‘some concerns’, or a ‘high risk of bias’ in accordance with the RoB 2 guidelines. The assessment was conducted independently by two evaluators, and in cases of disagreement, a third evaluator participated and made a final decision through discussion.

### 2.6. Statistical Analysis

All statistical analyses were performed using JASP (version 0.19.0.0) to account for the methodological and clinical heterogeneity between the studies. A random-effects model was applied using the Restricted Maximum Likelihood (REML) method to integrate the overall effect size, and the effect size was calculated as the Standardized Mean Difference (SMD) and 95% confidence interval (CI) [14].

The SMD was calculated according to the independent two-sample comparison (pretest–posttest–control group design) method using the pretest–posttest score difference [15,16]. More specifically, the mean difference between the groups in their posttest–pretest score differences was divided by the pooled standard deviation, accounting for covariance, and then multiplied by the Hedges’ g correction factor to reduce the bias in cases of a small sample size [17]. The standard error (SE) was calculated based on the sample size and effect size within each group [18]. The heterogeneity between the studies was quantitatively assessed using Cochran’s Q statistic, the I^2^ index, and τ^2^, and I^2^ was interpreted as being low, medium, or high at 25%, 50%, or 75%, respectively [12,19].

Subgroup analysis and meta-regression analysis were performed to explore the sources of heterogeneity and examine the impact of the moderator variables. The following variables were included as moderators in the meta-regression analysis: (1) the vibration type (whole-body vs. focal), (2) vibration frequency (Hz), (3) amplitude (mm), (4) vibration application posture, (5) total number of sessions, (6) stroke stage (acute, subacute, chronic), (7) type of assessment indicator (MAS, MTS, etc.). Each moderator variable was included in the meta-regression model as a categorical or continuous variable, and the Wald test and the omnibus test for model coefficients were performed to evaluate the reduction in the effect size’s heterogeneity and statistical significance. The regression model was fitted using the Restricted Maximum Likelihood (REML) method, and the model fit was compared using indices such as AIC, BIC, and AICc.

Publication bias was assessed through visual inspection of the funnel plot, Egger’s regression test, and Kendall’s tau rank correlation test, and Rosenthal’s fail-safe N was calculated to confirm the robustness of the analysis [20,21,22]. Sensitivity analysis was performed based on influence indicators such as the standardized residuals, Cook’s Distance, and the difference in the fits (DFFITS) to evaluate the impact of individual studies on the overall effect size estimate.

## 3. Results

### 3.1. Selection of Studies

A total of 1532 articles were identified through database searches (i.e., on PubMed, Cochrane, EMBASE, ScienceDirect, Web of Science), and 28 additional articles were identified through registries. Of these, 1499 duplicates were removed, and 61 articles were subsequently screened based on their titles and abstracts. As a result, 29 articles were excluded, and 32 articles were selected for a full-text review. Four articles were unavailable for retrieval, and twenty-eight articles were assessed for eligibility. Of these, 15 were excluded due to the following reasons: (1) unavailable outcome data (n = 6), (2) the absence of a control group (n = 5), (3) an inappropriate comparator intervention (n = 2), and (4) because they were conference abstracts or proceedings (n = 2). Ultimately, 13 studies were included in this systematic review and meta-analysis [23,24,25,26,27,28,29,30,31,32,33,34,35]. Meanwhile, seven additional records were identified using other methods (i.e., citation searching), of which five could not be retrieved. The remaining two records were excluded after eligibility assessment due to being non-English-language publications (n = 2); thus, no additional studies were included (Figure 1). 

### 3.2. Characteristics of the Studies

A total of 13 studies were included in this meta-analysis, of which 11 were randomized controlled trials (RCTs) and 2 were randomized crossover trials (RCOs) (Table 1). The studies mainly targeted patients with chronic or subacute stroke and applied vibration interventions to improve spasticity in the upper or lower extremities. The mean time since stroke onset for the patients included in the studies ranged from over 6 months to 102 months. Most studies focused on patients in the chronic phase (≥6 months), while only two studies involved participants in the acute or subacute phase [29,30].

The types of intervention were divided into FMV and WBV. A total of eight studies used FMV [25,27,28,29,30,31,32,33], and five studies used WBV [23,24,26,34,35].

The vibration parameters varied considerably across the studies: the frequency ranged from 4 Hz to 300 Hz, the amplitude ranged from 0.2 mm to 6 mm, the number of bouts per session ranged from 1 to 12, and the duration per bout ranged from 1 to 30 min, indicating substantial heterogeneity in the vibration dosage and stimulation protocols. Vibration was applied in standing, seated, or specific joint-fixed postures, and the equipment used included commercial devices such as Galileo and Viss, as well as custom-built stimulators developed by the research teams.

The effects of the interventions were primarily evaluated using the Modified Ashworth Scale (MAS), and some studies also used the Modified Tardieu Scale (MTS). Outcome assessments were conducted at various anatomical sites, including the fingers, wrists, elbows, shoulders, knees, and ankles. MAS/MTS scores were reported either for individual joints or as composite scores representing the upper or lower extremities.

### 3.3. Quality Assessment Results

A total of 13 randomized controlled trials (RCTs) were assessed for their risk of bias using Cochrane’s RoB 2.0 tool. Of these, two were based on a crossover design and 11 were based on a parallel-group design [36]. As shown in Figure 2, only three studies (23.1%) were assessed as having a low overall risk of bias, two studies (15.4%) were assessed as raising some concerns, and eight studies (61.5%) were assessed as having a high risk of bias. In other words, approximately 77% of the included studies were rated as raising at least some concerns or having a high risk of bias in one or more domains.

In particular, in the domain assessing the randomization process (Domain 1), many studies lacked adequate descriptions of their allocation concealment procedures, leading to assessments of a high risk or some concerns in this domain. In addition, in crossover design studies, the washout periods between the interventions were either not reported or insufficient, and the carryover effects were not statistically controlled, resulting in a high risk of bias in the domain related to period and carryover effects (Domain S). Furthermore, several studies were rated as high-risk in the domain assessing outcome measurement (Domain 4) because they used clinician-administered assessments such as the MAS and MTS, which are inherently subjective. In the absence of blinded outcome assessment, these measures may have been influenced by assessor expectations or bias, thereby increasing the risk of measurement bias. These findings suggest that in most studies, inadequate control of the randomization process and temporal design factors substantially contributed to the overall risk of bias.

### 3.4. The Effect of Vibration Interventiosn on Spasticity in Stroke Patients

Based on 34 comparison results from 13 randomized controlled trials (RCTs), the effect of vibration interventions on spasticity in stroke patients was analyzed. The results demonstrated that vibration interventions significantly reduced spasticity, with an overall average effect size of Hedges’ g = −0.50 (95% CI: −0.65 to −0.34, *p* < 0.001) (Figure 3). This represents a moderate magnitude of the effect, suggesting that vibration interventions may produce clinically meaningful reductions in spasticity.

In the heterogeneity analysis, the residual heterogeneity was Q = 66.38 (df = 33, *p* < 0.001), with an I^2^ value of 48.6% (95% CI: 27.1% to 76.7%), indicating a moderate level of between-study heterogeneity. This suggests that the variability in the effect sizes across the included studies was statistically significant and should be considered when interpreting the results.

When used to assess the risk of publication bias, both Kendall’s rank correlation test (τ = −0.574, *p* < 0.001) and Egger’s regression test (z = −5.112, *p* < 0.001) revealed significant funnel plot asymmetry, indicating potential publication bias (Figure 4).

However, the results of the Trim-and-Fill analysis showed that no additional studies were imputed, and the adjusted overall effect size remained −0.50 (95% CI: −0.65 to −0.34), consistent with the original estimate. This finding suggests that the spasticity-reducing effect of vibration interventions remained statistically robust even after accounting for potential publication bias, enhancing confidence in the reliability of the results.

Finally, the Rosenthal fail-safe N analysis indicated that 914 null studies would be required to render the observed effect non-significant, providing further support for the robustness and stability of the findings.

### 3.5. Meta-Regression Analysis of Moderators Influencing the Effects of Vibration on Spasticity in Stroke Patients

In this study, a meta-regression analysis was conducted to explore moderator variables that may influence the effects of vibration interventions on cognitive function. As a result, the amplitude, frequency, and time since stroke onset were identified as key moderators associated with the variation in the effect sizes (Table 2).

First, in the meta-regression model including the amplitude as a moderator, the overall model fit was not statistically significant (Q = 5.705, df = 2, *p* = 0.058). However, among the subgroups, the moderate-amplitude category (average amplitude of >0.5 mm and ≤1 mm) showed a significantly larger effect size than the low-amplitude reference group (≤0.5 mm) (β = 0.487, SE = 0.205, *p* = 0.018, 95% CI [0.085, 0.888]).

In the model including the frequency as a moderator, the overall model was also not statistically significant (Q = 5.353, df = 3, *p* = 0.148), but significant differences emerged in specific subgroups. The frequency was categorized into four levels: an average frequency > 100 Hz, >50 Hz and ≤100 Hz, >20Hz and ≤50 Hz, and ≤20 Hz. The 20–50 Hz group demonstrated a significantly greater effect size compared to the reference group (>100 Hz) (β = 0.518, SE = 0.246, *p* = 0.036, 95% CI [0.035, 1.001]). The 50–100 Hz group also showed a marginally significant difference relative to the reference group (β = 0.424, *p* = 0.053). In contrast, the ≤20 Hz group did not significantly differ from the reference (β = 0.258, *p* = 0.293).

The stroke onset timing was also a significant moderator of the effects of vibration interventions (Q = 17.273, df = 2, *p* < 0.001). The onset was grouped into three categories: the reference group included participants with an average onset ≤ 99 days ago (early subacute phase); the second group included those with an onset between 6 and 12 months ago (late subacute to early chronic phase); and the third group comprised individuals with onset ≥ 60 months ago (chronic phase). The second group exhibited a significantly smaller effect size compared to the reference (β = −0.606, SE = 0.199, *p* = 0.002, 95% CI [−0.996, −0.216]). No significant difference was found between the third group and the reference group (β = −0.053, *p* = 0.777).

In contrast, the vibration mode (e.g., vertical vs. oscillatory) did not significantly explain the variance in the effect sizes (Q = 0.007, df = 1, *p* = 0.934), and there was no significant difference between the categories (β = −0.015, *p* = 0.934). Additionally, no significant moderation effects were found for the total number of intervention sessions (Q = 2.346, df = 2, *p* = 0.309), daily vibration dosage (Q = 1.150, df = 1, *p* = 0.284), or outcome type (Q = 0.094, df = 1, *p* = 0.760).

### 3.6. Subgroup Analysis Based on Significant Moderators

Subgroup analyses were conducted for the amplitude, frequency, and stroke onset, which were identified as significant moderators in the meta-regression, to evaluate the differences in the effect sizes relative to those of the reference groups.

As a result of the subgroup analysis for the time since the stroke onset, the effect size was the largest in the group with an average onset date of more than 6 months but less than 12 months ago (late subacute to early chronic stage), at −0.87 (95% CI [−1.16, −0.58]), which was statistically significant. The chronic-stage group (60 months ago or more) also showed a significant effect size (−0.28, 95% CI [−0.43, −0.12]), but the early-stage group (99 days ago or less) had an effect size of −0.24 (95% CI [−0.52, 0.04]), which was not statistically significant. These results suggest that the period between 6 and 12 months post-stroke may represent an optimal time for maximizing the effects of vibration intervention (Figure 5).

In the amplitude subgroup analysis, low-amplitude stimulation of 0.5 mm or less showed the largest effect size (−0.83, 95% CI of [−1.26, −0.40]), high-amplitude stimulation of more than 1 mm showed an effect size of −0.52 (95% CI of [−0.78, −0.26]), and a medium amplitude of 0.5 to 1 mm showed an effect size of −0.26 (95% CI of [−0.43, −0.09]). All three groups showed statistically significant reductions in spasticity, and in particular, the fact that low-amplitude stimulation was the most effective indicates that a higher vibration intensity is not necessarily superior, but rather that better results can be expected at lower amplitudes (Figure 6).

In the frequency analyses, low-frequency stimulation below 20 Hz showed the largest effect size at −1.07 (95% CI of [−1.52, −0.62]). In addition, stimulation at 50–100 Hz showed an effect size of −0.40 (95% CI [−0.77, −0.04]), stimulation at 20–50 Hz showed an effect size of −0.22 (95% CI of [−0.37, −0.08]), and stimulation above 100 Hz showed an effect size of −0.78 (95% CI of [−1.16, −0.41]), and all four groups showed significant reductions in spasticity. In particular, the largest effect size was observed in the low-frequency condition below 20 Hz, followed by a relatively large effect in the high-frequency condition above 100 Hz. On the other hand, the effect size was relatively small in the middle frequency range (20–100 Hz). These results suggest that the pattern of the effect of the frequency may not simply be linear, but may show an inverted U-shaped or nonlinear trend (Figure 7).

These results demonstrate that the effect of vibration interventions can substantially vary depending on the timing of application, amplitude intensity, and frequency range. The most pronounced treatment benefits were observed when the intervention was delivered between 6 and 12 months post-stroke, using low-amplitude (≤0.5 mm) and very-low-frequency (<20 Hz) stimulation. A tailored intervention strategy that incorporates these optimal parameters may enhance the effectiveness of vibration-based rehabilitation in future clinical settings.

## 4. Discussion

This meta-analysis evaluated the overall efficacy of vibration interventions in reducing post-stroke spasticity and identified optimal application parameters through meta-regression and subgroup analyses. Synthesizing 13 randomized controlled trials (RCTs), the analysis revealed a statistically significant moderate effect size (Hedges’ g = −0.50; 95% CI: −0.65 to −0.34), confirmed by the fail-safe N and publication bias adjustment. These findings suggest that VT can produce clinically meaningful reductions in spasticity among stroke survivors.

Consistent with previous studies reporting the benefits of vibration for neuromuscular function and spasticity [24,25,27,28], this meta-analysis supports the use of both whole-body vibration (WBV) and focal muscle vibration (FMV) as viable rehabilitation strategies. Unlike earlier meta-analyses that combined heterogeneous outcomes (e.g., EMG, H-reflex, and clinical scales), which increased the statistical heterogeneity and reduced the clinical interpretability [19], the present study included only validated clinical scales (MAS, MTS), thereby improving the precision and clinical applicability of the pooled estimates.

Compared with previous key studies, our analysis shows notable methodological and interpretative differences. First, regarding the methodology, Elia et al. (2009) conducted a systematic review of the use of botulinum neurotoxins to treat post-stroke spasticity, including both randomized and non-randomized trials published up to 2008, without restricting the outcome measures to a specific clinical scale [2]. In contrast, our study included only randomized controlled trials and restricted the spasticity outcomes to validated measures (MAS and MTS) to improve the data homogeneity. We also employed meta-regression and subgroup analyses to examine parameter-specific effects, which were not addressed in Elia et al.’s review [4]. Pang et al. (2013) [35], on the other hand, conducted a single-center randomized controlled trial on whole-body vibration therapy in chronic stroke patients, focusing on a broad set of outcomes including bone turnover, muscle strength, motor function, and spasticity. Our study differs in scope by synthesizing evidence across multiple trials and interventions and by specifically targeting vibration therapy parameters as moderators.

Second, regarding the results, Elia et al. reported consistent spasticity reduction across most studies but did not quantify the heterogeneity or identify moderators [2]. Our findings indicate a moderate pooled effect with substantial heterogeneity, partially explained by differences in vibration parameters such as the frequency and time since stroke onset. Pang et al. observed modest improvements in spasticity and motor outcomes following whole-body vibration therapy, though not all the measures reached statistical significance. In our meta-analysis, the effect sizes were larger when interventions were applied in the late subacute to early chronic stage, suggesting time-sensitive efficacy that was not explicitly analyzed in Pang et al.’s trial [35].

Third, in terms of interpretation, Elia et al. emphasized the established clinical role of botulinum neurotoxins as a first-line pharmacological treatment, whereas our study highlights the potential of vibration therapy for use as a non-pharmacological neuromodulatory option [2]. Pang et al. interpreted their findings cautiously, noting limited generalizability due to their small sample size and single-intervention protocol [35]. Our results extend this perspective by demonstrating that heterogeneity in the vibration parameters may account for divergent findings across studies, underscoring the need for individualized parameter optimization in clinical practice.

The physiological rationale for VT is well-supported. Vibration stimulates mechanoreceptors and proprioceptive afferents, such as muscle spindles and Golgi tendon organs, leading to modulation of inhibitory spinal reflex circuits and suppression of alpha motor neuron hyperexcitability. These effects are thought to reduce muscle tone and promote spinal–cortical reorganization and improved sensorimotor integration [28,37]. Additionally, vibration induces micro-contractions that enhance local circulation and joint flexibility. FMV, in particular, can modulate motor unit recruitment and strengthen corticospinal connectivity through targeted stimulation [25]. Collectively, these mechanisms support the use of VT as a neuromodulatory approach to managing post-stroke spasticity.

Meta-regression and subgroup analyses identified the vibration frequency, amplitude, and time since stroke onset as key moderators of the treatment efficacy. Notably, applying vibration between 6 and 12 months post-stroke yielded the largest effect size (g = −0.87), suggesting that this period may represent an underrecognized therapeutic window for maximizing the benefits of neuromodulatory interventions. While this timeframe does not fully align with the widely cited 3–6-month window of peak neuroplasticity [38,39,40], it is likely that substantial plastic potential persists beyond the early subacute stage. During this period, many patients experience relative stabilization of their motor and cognitive status following initial recovery, which may facilitate better engagement with vibration-based interventions. Moreover, vibration therapy provides low-effort somatosensory stimulation capable of activating motor pathways without inducing fatigue, potentially improving adherence and functional gains in those with reduced activity tolerance. These factors, together with the persistence of substantial neuroplastic potential beyond the early subacute stage [41], may collectively explain the heightened responsiveness to vibration therapy observed between 6 and 12 months post-stroke, highlighting the value of tailoring interventions to specific recovery phases [7].

Contrary to the assumption that greater intensity leads to better outcomes, low-amplitude vibration (≤0.5 mm) demonstrated the greatest treatment effect (g = −0.83). This may have been due to reduced peripheral fatigue, improved patient comfort, and effective stimulation of mechanoreceptors at lower thresholds. Pacinian corpuscles and muscle spindles are known to respond to small-amplitude stimuli [42,43], and such stimulation may facilitate sustained activation without fatigue [7]. These findings support the concept that less intense, more precisely calibrated stimuli can be more effective than high-intensity inputs.

Frequency analysis revealed a nonlinear relationship, where both low (<20 Hz) and high (>100 Hz) frequencies were more effective than mid-range frequencies, suggesting a U-shaped or bi-modal response pattern. These insights underscore the need for individualized parameter selection rather than a one-size-fits-all approach.

Despite variability in the vibration dosage, session duration, body posture, and device type (e.g., Galileo, Viss, and custom devices), the effect sizes remained robust, indicating the generalizability of VT across diverse clinical settings. Although most included studies targeted chronic stroke patients, the most pronounced effects were observed in those in the late subacute to early chronic phase, further supporting the need for earlier intervention.

Several limitations should be acknowledged. First, the variability in the methodological quality, including unclear randomization, inadequate allocation concealment, and insufficient washout periods in crossover designs, increased the risk of bias [13]. In addition, the overall heterogeneity (I^2^) observed in the pooled analysis was substantial, stemming from differences in the participant characteristics, vibration parameters, intervention durations, and outcome assessment methods. While subgroup and meta-regression analyses were able to explain part of this variability, residual heterogeneity remained, which may limit the generalizability of the effect size estimates. This unexplained heterogeneity suggests that unmeasured clinical or methodological factors—such as therapist expertise, patient adherence, or comorbidities—may have influenced the treatment responses. Second, the small sample sizes in many trials may have reduced the power to detect moderator effects. Third, the exclusion of non-English language studies, despite adjustments for potential publication bias, may have introduced a language bias, thereby limiting the comprehensiveness and generalizability of the findings.

Nevertheless, this meta-analysis reinforces the potential of VT for use as a non-invasive, cost-effective, and accessible intervention for post-stroke spasticity. In particular, low-frequency, low-amplitude vibration applied during the 6–12-month post-stroke period appears most beneficial. Future research should focus on several key directions. First, harmonization and standardization of vibration therapy protocols—covering parameters such as the frequency, amplitude, session duration, and treatment schedule—are needed to enable more reliable cross-study comparisons and facilitate clinical implementation. Second, future trials should incorporate broader functional outcomes, including gait performance, balance, and activities of daily living (ADL), to determine whether spasticity reduction translates into meaningful functional improvements. Third, mechanistic investigations are warranted to clarify the neurophysiological pathways underlying vibration therapy’s effects, such as spinal reflex modulation, corticospinal excitability, sensory–motor integration, and neuroplastic changes across recovery phases. Employing neuroimaging, electrophysiological assessments, and biomarker analyses will provide stronger objective evidence to guide optimized intervention strategies. By addressing these areas, subsequent research can refine clinical protocols, enhance patient selection, and maximize the therapeutic potential of vibration therapy in post-stroke rehabilitation.

## 5. Conclusions

This meta-analysis quantitatively demonstrated that VT is effective in reducing spasticity following stroke. The pooled effect size was moderate (Hedges’ g = −0.50), indicating a clinically meaningful reduction in muscle tone. Subgroup analyses revealed that the most pronounced effects were achieved when the intervention was applied during the late subacute to early chronic stage (6–12 months post-stroke) and when low-frequency (<20 Hz) and low-amplitude (≤0.5 mm) stimulation was used. These findings suggest that VT operates through neuromodulatory mechanisms that are sensitive to the timing and stimulation intensity.

Among the seven moderator variables tested, the time since stroke onset, vibration frequency, and amplitude significantly influenced the treatment outcomes. Conversely, the posture, total number of sessions, vibration type (WBV vs. FMV), and outcome measure (MAS vs. MTS) showed no significant moderating effects. This indicates that the therapeutic efficacy of vibration is more dependent on the intensity and timing than on the duration or delivery mode.

This study also adopted rigorous methodological approaches, including calculation of effect sizes based on pre–post changes, risk-of-bias assessment, and correction for publication bias. However, its limitations include small sample sizes, variation in the intervention protocols, and insufficient mechanistic data.

Nonetheless, VT offers a promising, non-invasive alternative for stroke survivors with spasticity, especially for those unable to tolerate intensive physical therapy. Future studies should explore the neurophysiological mechanisms underlying vibration-induced spasticity reduction and aim to identify optimal patient-specific stimulation parameters, enabling the development of personalized rehabilitation protocols.

## Figures and Tables

**Figure 1 jcm-14-05966-f001:**
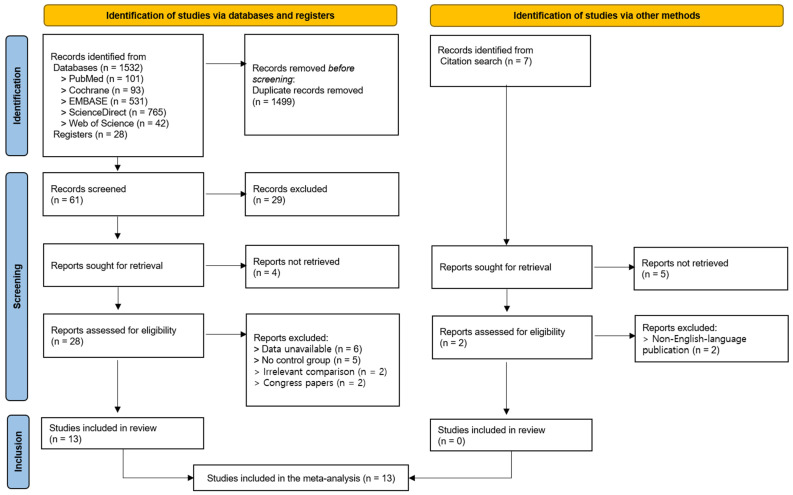
Flowchart of study selection.

**Figure 2 jcm-14-05966-f002:**
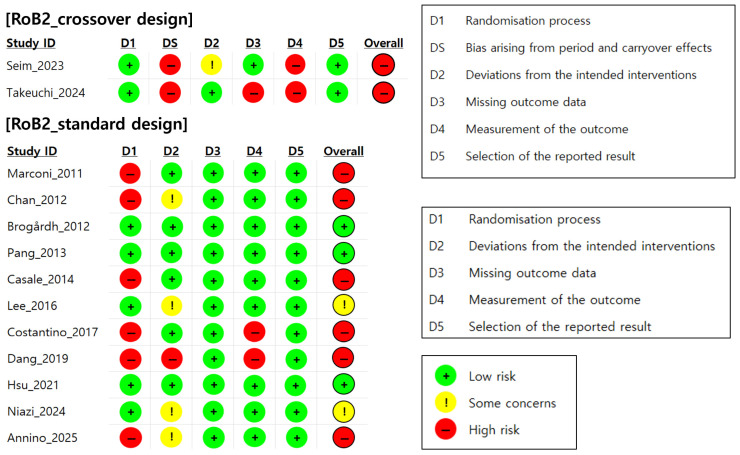
Quality assessment results by risk of bias [23,24,25,26,27,28,29,30,31,32,33,34,35].

**Figure 3 jcm-14-05966-f003:**
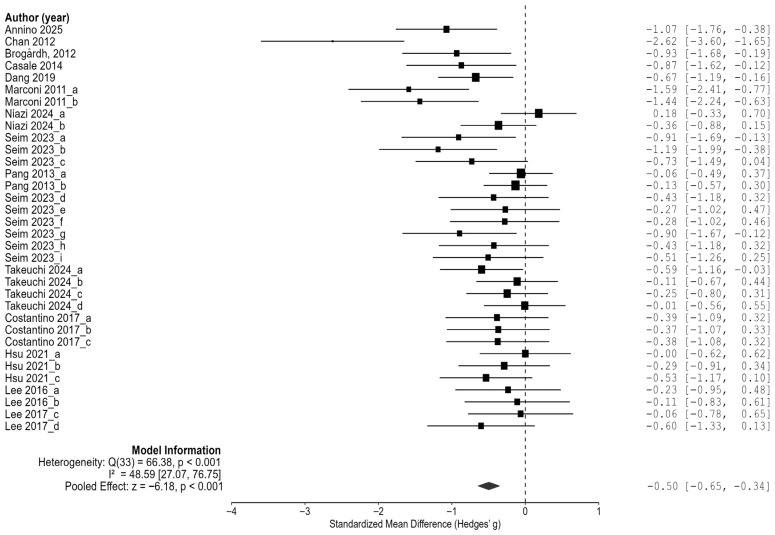
The effect of vibration interventions on spasticity in stroke patients [23,24,25,26,27,28,29,30,31,32,33,34,35].

**Figure 4 jcm-14-05966-f004:**
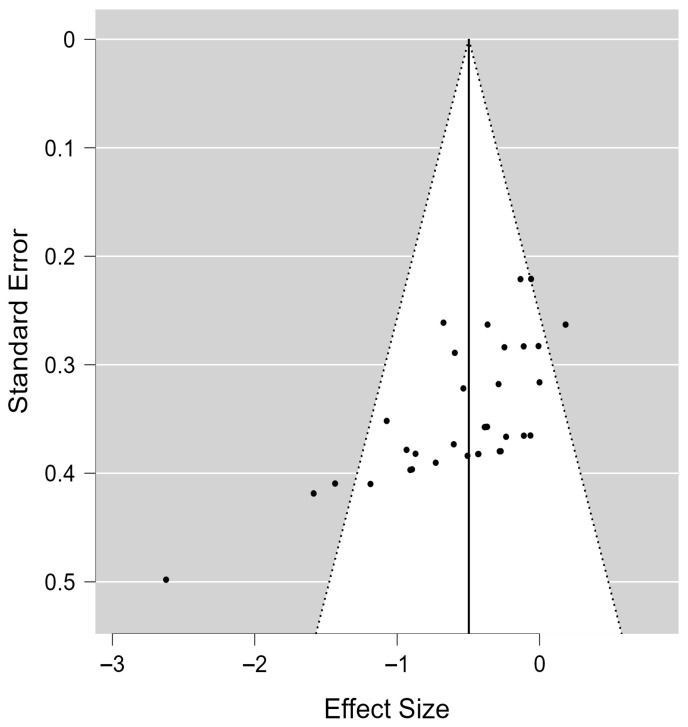
The funnel plot of the included studies.

**Figure 5 jcm-14-05966-f005:**
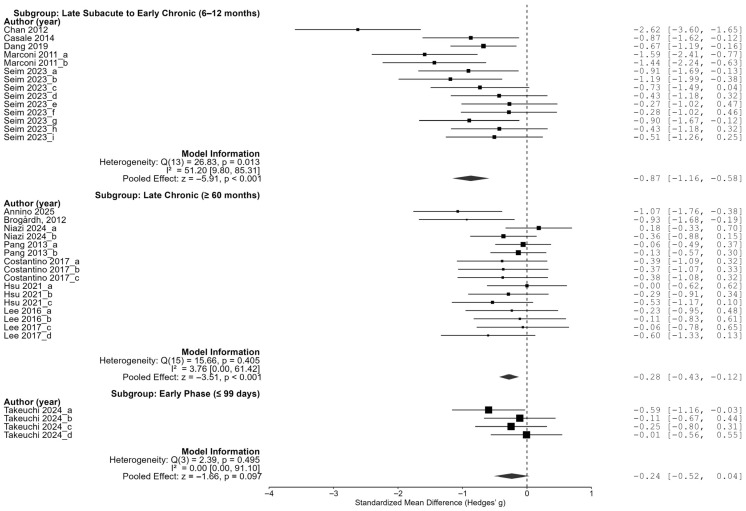
Subgroup analysis by time since stroke onset [23,24,25,26,27,28,29,30,31,32,33,34,35].

**Figure 6 jcm-14-05966-f006:**
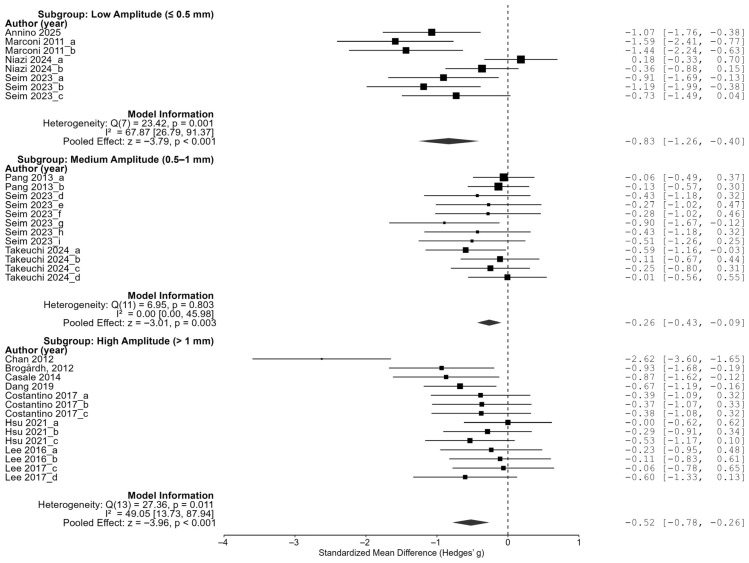
Subgroup analysis by amplitude [23,24,25,26,27,28,29,30,31,32,33,34,35].

**Figure 7 jcm-14-05966-f007:**
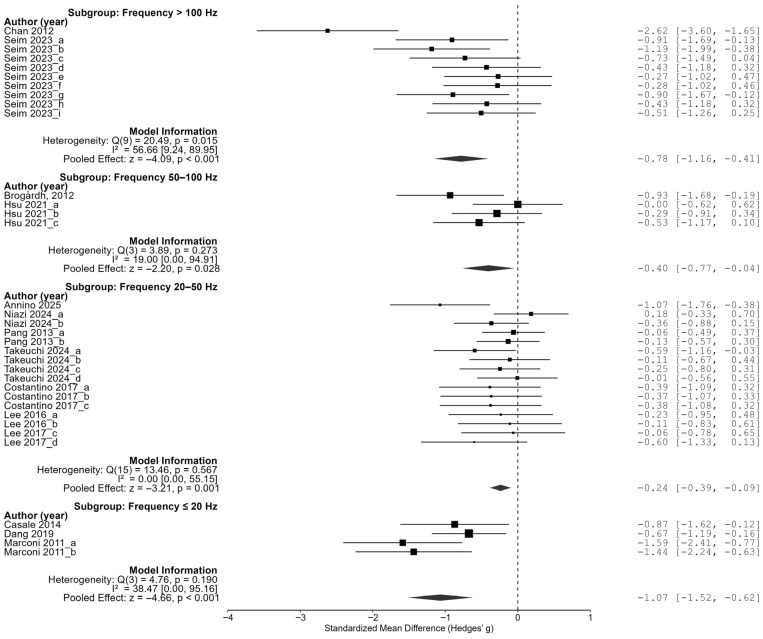
Subgroup analysis by vibration frequency [23,24,25,26,27,28,29,30,31,32,33,34,35].

**Table 1 jcm-14-05966-t001:** Study characteristics for WBV interventions.

Author and Year	Study Type	Participants	Post-Stroke Duration (Mean)	Sample (N, Age)	Intervention and Control Protocol	Intervention Protocol	Vibration Parameters (Frequency, Amplitude)	Vibration Dose Parameters (Bouts/Session, Dur/Bout (Min))	Posture and WBV Type	Device	Outcome Measures (Measurement Site)
Annino et al., 2025 [29]	RCT	Chronic stroke	N.A.	I: 19 (67.8 ± 8.3); C: 18 (69.4 ± 10.4)	FMV	Freq: 3/week; Dur: 8 wk	30 Hz, 0.2 mm	3/session, 10 min	Quadriceps tendon region	FOV1 Power Club (Ferrara, Italy)	MAS (knee)
Brogårdh et al., 2012 [23]	RCT	Chronic stroke patients	72 months	I: 16 (61.3 ± 8.5); C: 15 (63.9 ± 5.8)	WBV	Freq: 2/week; Dur: 6 wk	25 Hz, 3.75 mm	12/session, 1 min	Static standing, knees flexed 45–60°	Xrsize (Askim, Sweden	MAS (muscle tone of lower limb)
Casale et al., 2014 [25]	RCT	Adults with chronic stroke	12 months	I: 15 (64.7 ± 5.4); C: 15 (65.1 ± 5.8)	FMV	Freq: 5/week; Dur: 2 wk	100 Hz, 2 mm	1/session, 30 min	Seated position with vibration applied to biceps and triceps brachii	VIBRA (Circle, Ferrara, Italy)	MAS (biceps brachii)
Chan et al., 2012 [24]	RCT	Chronic stroke patients	≥6 months	I: 15 (56.07); C: 15 (54.93)	WBV	Freq: 1/week; Dur: 1 wk	12 Hz, 4 mm	2/session, 10 min	Semi-squat with buttock support	AV-001 (Bodygreen, Xiushui Township, Taiwan)	MAS (ankle)
Costantino et al., 2017 [31]	RCT	Chronic upper limb spasticity	37 months	I: 17 (62.59 ± 15.39); C: 15 (60.47 ± 16.09)	FMV	Freq: 3/week; Dur: 4 wk	300 Hz, 2 mm	1/session, 30 min	Local mechano-acoustic vibration applied to triceps and extensor carpi radialis (SP2) during isometric contraction	Viss^®^ device (Vissman, Rome, Italy)	MAS (elbow, shoulder, wrist)
Dang et al., 2019 [34]	RCT	Post-stroke patients with upper limb hemiplegia	≥6 months	I1: 32 (57.8 ± 8.5); I2: 33 (60.6 ± 8.0); C: 30 (60.0 ± 9.0)	WBV	Freq: 5/week; Dur: 4 wk	5–15 Hz, 1–6 mm	I1: 1/session, 30 min I2: 1/session, 30 min	Sitting, palms on vibration platform at 90° shoulder flexion	SVG (WellenGang GmbH, Mühlacker, Germany)	MAS (upper limb)
Hsu et al., 2021 [32]	RCT	Chronic stroke	29 months	I: 20 (57.3 ± 6.5); C: 20 (59.1 ± 5.8)	FMV	Freq: 2/week; Dur: 6 wk	30 Hz, 2 mm	8/session, 2 min	Seated posture, hand grasping pinch-force device	Custom-built pinch vibration stimulator (Tainan, Taiwan)	MAS (elbow, wrist, fingers)
Lee et al., 2016 [26]	RCT	Patients with chronic hemiplegia after stroke	33 months	I1: 15 (59.20 ± 7.72); I2: 15 (58.53 ± 11.83); C: 15 (60.24 ± 6.73)	I1: WBV I2: WBV + TR	Freq: 3/week; Dur: 4 wk	4–15 Hz, 2–6 mm	7/session, 2 min	Seated on chair, both shoulders flexed 90º, trunk bent forward, palms on vibration platform	Galileo tilting table (Novotec Medical GmbH (Pforzheim, Germany)	MAS (total, elbow, shoulder, wrist/finger)
Marconi et al., 2011 [28]	RCT	Stroke patients (cortical or subcortical lesion)	11 months	I: 15 (63.6 ± 7.6); C: 15 (66.3 ± 11.0)	FMV	Freq: 5/week; Dur: 2 wk	100 Hz, 0.2–0.5 mm	3/session, 10 min	Over target muscle (FCR, BB, EDC)	Cro®System (NEMICI S.r.L.; Rome, Italy)	MAS (elbow, wrist)
Niazi et al., 2024 [33]	RCT	Subacute stroke patients	102 months	I1: 30 (50.53 ± 11.13); I2: 30 (50.56 ± 10.36); C: 30 (51 ± 6.67)	FMV	Freq: 3/week; Dur: 8 wk	I1: 60 Hz, 0.5 mm I2: 120 Hz, 0.5 mm	1/session, 10 min	Biceps brachii and extensor carpi radialis of paretic limb	Custom-developed wearable FMV sleeve device (Auckland, New Zealand)	MAS (biceps brachii and extensor carpi radialis of paretic limb)
Pang et al., 2013 [35]	RCT	Chronic stroke patients	60 months	I: 41 (57.3 ± 11.3); C: 41 (57.4 ± 11.1)	WBV	Freq: 3/week; Dur: 8 wk	20–30 Hz, 0.44–0.60 mm	6/session, 1.5–2.5 min	Standing, semi-squat, lunge, single leg, deep squat	N.A.	MAS (ankle, knee)
Seim et al., 2023 [27]	RCO	Chronic stroke patients	12 months	I: 14 (60 ± 10.5); C: 14 (60 ± 10.5)	FMV (agonist muscle, antagonist muscle, finger cutaneous)	Freq: 1/week; Dur: 4 wk	I1: 67–70 Hz, 0.61 mm I2: 253–255 Hz, <0.08 mm	1/session, 20 min	Agonist muscle stimulation	C08-007 (Precision Microdrives Limited, London, UK)	MTS (fingers)
Takeuchi et al., 2024 [30]	RCO	Post-stroke upper extremity spasticity	99 days	I: 25 (61.3 ± 12.4); C: 25 (61.3 ± 12.4)	FMV (tendon, muscle belly)	Freq: 3/week; Dur: 1 wk	91 Hz, 1 mm	1/session, 5 min	Vibratory stimulation for tendon	Thrive MD-01 (Thrive Co. Ltd., Osaka, Japan)	MAS (fingers, wrist)

Abbreviations: I, intervention group; C, control group; RCT, randomized controlled trial; RCO, randomized crossover trial; WBV, whole-body vibration; FMV, focal muscle vibration; MAS, Modified Ashworth Scale; MTS, Modified Tardieu Scale; Freq, frequency; Dur, duration; wk, weeks; min, minutes; mm, millimeters; Hz, Hertz.

**Table 2 jcm-14-05966-t002:** Results of meta-regression analyses of effect of moderator variables on antispasticity effects of vibration interventions.

Moderator Variable	Level	Reference Level	β	95% CI	*z*	*p*-Value	Omnibus Q (df)	Omnibus *p*
Amplitude	>0.5 mm and ≤1 mm	≤0.5 mm	0.487	[0.085, 0.888]	2.374	0.018 *	5.705 (2)	0.058
	>1 mm		0.266	[−0.134, 0.666]	1.305	0.192		
Stroke onset	≥6 months and <12 months ago	≤99 days ago	−0.606	[−0.996, −0.216]	−3.047	0.002 *	17.273 (2)	<0.001 *
	≥60 months ago		−0.053	[−0.420, 0.314]	−0.283	0.777		
Frequency	>50 Hz and ≤100 Hz	>100 Hz	0.424	[−0.005, 0.853]	1.936	0.053	5.353 (3)	0.148
	>20 Hz and ≤50 Hz		0.518	[0.035, 1.001]	2.102	0.036 *		
	≤20 Hz		0.258	[−0.222, 0.739]	1.053	0.293		
Total number of sessions	6–19 sessions	≤5 sessions	0.27	[−0.082, 0.623]	1.505	0.132	2.346 (2)	0.309
	≥20 sessions		0.107	[−0.358, 0.572]	0.452	0.652		
Vibration type	FMV	WBV	−0.015	[−0.375, 0.345]	−0.083	0.934	0.007 (1)	0.934
Daily vibration dosage	≥6 min	≤5 min	0.17	[−0.141, 0.482]	1.072	0.284	1.150 (1)	0.284
Outcome type	MTS	MAS	−0.093	[−0.692, 0.505]	−0.306	0.76	0.094 (1)	0.76

Regression coefficients (β), 95% confidence intervals (CI), z-values, and *p*-values are reported for each level of the moderator variables, along with the reference level. Omnibus Q tests indicate whether the between-group differences in the effect sizes were statistically significant. *p*  <  0.05 is considered statistically significant and is indicated with an asterisk (*). Total session number was calculated as the number of sessions per week multiplied by the number of weeks the intervention was administered for (sessions/week × weeks). Daily vibration dosage was calculated as the number of vibration bouts per session multiplied by the duration of each bout in minutes (bouts × minutes). Abbreviations: CI, confidence interval; FMV, focal muscle vibration; WBV, whole-body vibration; MTS, Modified Tardieu Scale; MAS. Modified Ashworth Scale.

## Data Availability

The data supporting the findings of this study are available from the corresponding author upon reasonable request.

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
