# Peer review of "The Effects of Whole-Body Vibration on Spasticity in Stroke: A Systematic Review and Meta-Analysis"

_jcm, 2025, doi:10.3390/jcm14175966_

Round 1
Reviewer 1 Report
Comments and Suggestions for Authors
It’s a great honor for me to read such a topic.
The present manuscript presents a focused and methodologically detailed systematic review and meta-analysis on the effects of vibration therapy (VT) in reducing post-stroke spasticity. The choice of research is in a field of great interest recently due to the need of more and more clearness in the rehab.
The title accurately reflects the study’s scope. The abstract is very well-structured, clearly summarizing the rationale, methods, key findings, and conclusions.
The introduction provides a comprehensive overview of the clinical burden of spasticity after stroke and the limitations of pharmacological and physical rehabilitation modalities, offering a strong rationale for VT. However, the novelty of this review is not clearly distinguished from previous meta-analyses; a more explicit articulation of the study’s unique contribution—such as advanced meta-regression modeling or a stricter focus on MAS and MTS outcomes—is warranted.
The methods section is good and follows thee PRISMA guidelines. The inclusion/exclusion criteria are appropriate and well defined, and the study employs rigorous quality assessment using the RoB 2 tool. Nevertheless, considerable clinical heterogeneity in VT protocols—variability in frequency (4–300 Hz), amplitude (0.2–6 mm), delivery mode (WBV vs. FMV), posture, and dosage—is only partially addressed in subgroup analysis and meta-regression. These variations may have a substantial impact on outcomes but are not sufficiently explored or discussed in terms of their influence on pooled estimates. Moreover, the exclusion of non-English studies limits comprehensiveness and may introduce language bias, a limitation underemphasized in the manuscript. The statistical analysis is a strength, with appropriate use of pretest–posttest control group designs, Hedges’ g correction for small samples, and calculation of heterogeneity (I² = 48.6%) alongside sensitivity analysis (funnel plots, Egger’s test, fail-safe N). Moderator analyses identify frequency, amplitude, and stroke onset as significant variables, and the subgroup analyses yield clinically meaningful trends, particularly that low-frequency (<20 Hz), low-amplitude (≤0.5 mm) VT applied in the 6–12-month post-stroke window is most effective. This is a valuable finding, though the mechanistic rationale for this timing window—deviating from the traditional 3–6-month neuroplastic period—could be better explained physiologically. The results are clearly presented and well-supported by quantitative synthesis, yet the implications of the high risk of bias (only 23.1% of included studies rated low risk) and publication bias (evident funnel plot asymmetry) are insufficiently emphasized. The discussion correctly aligns with the findings, integrates prior literature, and explains neuromodulators mechanisms (e.g., proprioceptive feedback, spinal reflex suppression), but it would benefit from a more cautious interpretation of effect sizes and a call for standardization of VT parameters. Moreover, comparisons with other systematic reviews (e.g., Elia et al., Pang et al.) are too superficial and fail to clarify this study’s added value.
The conclusions are mostly justified by the data, but phrasing such as “optimal parameters” should be moderated given the moderate effect size, residual heterogeneity, and risk of bias. Figures and tables are clear and generally well-presented, although some subgroup analyses would benefit from improved axis labeling and more detailed captions to enhance interpretability.
Overall, the manuscript makes a valuable contribution by identifying meaningful VT parameters and suggesting personalized protocols in stroke rehabilitation, but requires improvement in discussing methodological variability, bias, and positioning relative to the existing literature.
I would recommend if the authors decide those improvements are feasible according to their deeper experience:
• To clarify the novelty of this meta-analysis compared to previous reviews, especially regarding methodology, scope, and clinical impact.
• Discuss in greater depth the implications of high heterogeneity and risk of bias, and consider sensitivity analysis excluding high-risk studies. This is well known difficultly in meta-analyses and such articles, but still it should be addressed in detail.
• Include in limitations the possible language bias from the exclusion of non-English studies and explicitly state it as a limitation.
• Improve discussion with comparison with results from other recent meta-analyses (e.g., Elia et al., Pang et al.), highlighting differences in methods, outcomes, or conclusions.
• If applicable it would be interesting to provide stronger physiological justification for the efficacy of VT during the 6–12-month post-stroke window and address whether this window is supported by neuroplasticity literature.
• Standardize figure presentation, especially in subgroup analyses, by improving axis labels and adding interpretive figure captions, where possible.
• In the end you can give a recommendation for future research directions like more strongly, including protocol harmonization, functional outcome inclusion (e.g., gait, ADLs), and mechanistic studies of VT effects. This will give a path for other researchers to continue the work in the field.
Author Response
Comments 1: To clarify the novelty of this meta-analysis compared to previous reviews, especially regarding methodology, scope, and clinical impact
Response 1: We appreciate this valuable suggestion. We have expanded the Introduction to more explicitly differentiate our work from prior meta-analyses. (Line 83-116)
Comments 2: Discuss in greater depth the implications of high heterogeneity and risk of bias, and consider sensitivity analysis excluding high-risk studies. This is well known difficultly in meta-analyses and such articles, but still it should be addressed in detail.
Response 2: The limitations of this study were added to the last part of the discussion as follows. “In addition, the overall heterogeneity (I²) observed in the pooled analysis was substantial, stemming from differences in participant characteristics, vibration parameters, intervention durations, and outcome assessment methods. While subgroup and meta-regression analyses were able to explain part of this variability, residual heterogeneity remained, which may limit the generalizability of the effect size estimates. This unexplained heterogeneity suggests that unmeasured clinical or methodological factors—such as therapist expertise, patient adherence, or comorbidities—may have influenced treatment responses.” (Line 525-532)
Comments 3: Include in limitations the possible language bias from the exclusion of non-English studies and explicitly state it as a limitation.
Response 3: The limitations of this study were added to the last part of the discussion as follows. “Third, the exclusion of non-English language studies, despite adjustments for potential publication bias, may have introduced a language bias, thereby limiting the comprehensiveness and generalizability of the findings.” (Line 534-536)
Comments 4: Improve discussion with comparison with results from other recent meta-analyses (e.g., Elia et al., Pang et al.), highlighting differences in methods, outcomes, or conclusions.
Response 4: The revised content through comparison with previous studies has been added to the Discussion section.
“Compared with previous key studies, our analysis shows notable methodological and interpretative differences. First, in methodology, Elia et al. (2009) conducted a systematic review of botulinum neurotoxins for post-stroke spasticity, including both randomized and non-randomized trials published up to 2008, without restricting outcome measures to a specific clinical scale. In contrast, our study included only randomized controlled trials and restricted spasticity outcomes to validated measures (MAS and MTS) to improve data homogeneity. We also employed meta-regression and subgroup analyses to examine parameter-specific effects, which were not addressed in Elia et al.’s review [4]. Pang et al. (2013) [27], on the other hand, conducted a single-center randomized controlled trial on whole-body vibration therapy in chronic stroke patients, focusing on a broad set of outcomes including bone turnover, muscle strength, motor function, and spasticity. Our study differs in scope by synthesizing evidence across multiple trials and interventions, and by specifically targeting vibration therapy parameters as moderators.
Second, regarding results, Elia et al. reported consistent spasticity reduction across most studies but did not quantify heterogeneity or identify moderators [4]. Our findings indicate a moderate pooled effect with substantial heterogeneity, partially explained by vibration parameters such as frequency and time since stroke onset. Pang et al. observed modest improvements in spasticity and motor outcomes following whole-body vibration therapy, though not all measures reached statistical significance. In our meta-analysis, effect sizes were larger when interventions were applied in the late subacute to early chronic stage, suggesting time-sensitive efficacy that was not explicitly analyzed in Pang et al.’s trial [27].
Third, in interpretation, Elia et al. emphasized the established clinical role of botulinum neurotoxins as a first-line pharmacological treatment, whereas our study highlights the potential of vibration therapy as a non-pharmacological neuromodulatory option [4]. Pang et al. interpreted their findings cautiously, noting limited generalizability due to their small sample size and single intervention protocol [27]. Our results extend this perspective by demonstrating that heterogeneity in vibration parameters may account for divergent findings across studies, underscoring the need for individualized parameter optimization in clinical practice.” (Line 451-480)
Comments 5: If applicable it would be interesting to provide stronger physiological justification for the efficacy of VT during the 6–12-month post-stroke window and address whether this window is supported by neuroplasticity literature.
Response 5: I added a reference [40] on the importance of early stages in neuroplasticity at Discussion. I also rewrote the content for clarity as follows.
“These factors, together with the persistence of substantial neuroplastic potential beyond the early subacute stage [40], may collectively explain the heightened responsiveness to vibration therapy observed between 6 and 12 months post-stroke, highlighting the value of tailoring interventions to specific recovery phases [7].” (Line 501-505)
Comments 6: Standardize figure presentation, especially in subgroup analyses, by improving axis labels and adding interpretive figure captions, where possible.
Response 6: Based on your feedback, we have supplemented the axis labels and descriptive captions in the subgroup analyses in Figures 3, 5, 6, and 7.
Comments 7: In the end you can give a recommendation for future research directions like more strongly, including protocol harmonization, functional outcome inclusion (e.g., gait, ADLs), and mechanistic studies of VT effects. This will give a path for other researchers to continue the work in the field.
Response 7: Based on your feedback, I've added this to the end of the discussion. The content is as follows:
“Nevertheless, this meta-analysis reinforces the potential of VT as a non-invasive, cost-effective, and accessible intervention for post-stroke spasticity. In particular, low-frequency, low-amplitude vibration applied during the 6–12 month post-stroke period appears most beneficial. Future research should focus on several key directions. First, harmonization and standardization of vibration therapy protocols—covering parameters such as frequency, amplitude, session duration, and treatment schedule—are needed to enable more reliable cross-study comparisons and facilitate clinical implementation. Second, future trials should incorporate broader functional outcomes, including gait performance, balance, and activities of daily living (ADL), to determine whether spasticity reduction translates into meaningful functional improvements. Third, mechanistic investigations are warranted to clarify the neurophysiological pathways underlying vibration therapy’s effects, such as spinal reflex modulation, corticospinal excitability, sensory-motor integration, and neuroplastic changes across recovery phases. Employing neuroimaging, electrophysiological assessments, and biomarker analyses will provide stronger objective evidence to guide optimized intervention strategies. By addressing these areas, subsequent research can refine clinical protocols, enhance patient selection, and maximize the therapeutic potential of vibration therapy in post-stroke rehabilitation.” (Line 537-553)
Reviewer 2 Report
Comments and Suggestions for Authors
Comments
The authors conducted a systematic review and meta-analysis to examine the impact of whole-body vibration on spasticity. However, despite its relevance, several methodological and procedural issues must be addressed.
- Abstract: “Thirteen randomized controlled trials (RCTs) involving whole-body”: This should be in the results section (of the abstract). In the Methods section, you only need to specify which studies were included, not how many.
- Introduction: Although it's generally good, I would suggest making a few improvements.
- Introduction: For example, it's important not to leave things uncited. For example, lines 41 to 48, to name just a few.
- Introduction: Lines 63–64: paragraphs of only two lines are strongly discouraged.
- Introduction: While it is true that you should report on what was unknown in previous meta-analyses in order to justify yours, this should be summarised in no more than six lines. There's no need to explain in depth what other authors did and what they did wrong – just summarise what they're missing and what you're going to address.
- Methods: In addition to PRISMA, authors should follow and cite the Cochrane Collaboration Handbook.
- Methods: The PRISMA Checklist should be included in the supplementary material.
- Methods: “2.1. Study Design”: Lines 119-124: I don't understand why these lines are necessary, as they simply reiterate the objective and summarise the inclusion criteria, which are included later in the manuscript.
- Methods: “The search language was limited to English.” With current translation tools (not including AI), there is little justification for limiting oneself to English, even though it is the dominant language in science.
- Methods: “The inclusion of studies was independently assessed by two or more reviewers, and 141 disagreements were resolved through discussion or consultation with a third reviewer.”: two or more reviewers? Either there are two, or there are three, or there are… a specific number. Specify this, and who these two reviewers are. I also suggest changing the term "reviewer" to "author" (because I imagine the authors of the manuscript performed these procedures).
- Methods: Similarly, other sections should specify which authors performed each procedure (e.g. risk of bias assessment, data extraction) and their initials.
- Methods: The use of GRADE to assess the strength or quality of the evidence would enrich this systematic review and meta-analysis.
- Methods: “The random-effects model 220 was applied to integrate the overall effect size, and the effect size was calculated as Stand-221 ardized Mean Difference (SMD) and 95% confidence interval (CI) [14].”: What type of random effects were used? Dersimonian and Laird? Or which one?
- Results: The PRISMA and AMSTAR-2 checklists require a list of excluded studies with the main reason for exclusion for each study. Include it in supplementary material.
- Results: “Ultimately, 13 studies were included in this systematic review and meta-analysis.”: Authors should include citations here, as this is the first time they refer to all 13 included studies.
- Results: Some of the studies included in the systematic review (references 1, 2, 3, 5, 6, 8) are used in the introduction. The studies included in the systematic review should never be included in the introduction. Occasionally, if imperative, it can be used to explain the need for your study. But, in general, it should not be used. I mean, for example, and this is just an example: “Spasticity is a frequent and debilitating sequela of stroke, characterized by a velocity-dependent increase in muscle tone due to hyperexcitability of the stretch reflex. This pathophysiological condition impairs voluntary movement, restricts joint mobility, and hinders functional recovery, thereby markedly diminishing patients’ quality of life.” You cannot use reference 1, which is a clinical trial that aimed to analyse the efficacy of an intervention. Appropriate references should be used, considering the objective of the referenced study. I don’t know if I’m making myself clear.
- Discussion: In my opinion, there's nothing more I can add — it seems coherent and complete.
Author Response
Comments 1: Abstract: “Thirteen randomized controlled trials (RCTs) involving whole-body”: This should be in the results section (of the abstract). In the Methods section, you only need to specify which studies were included, not how many.
Response 1: Based on your feedback, we have moved to the results section.
Comments 2: Introduction: For example, it's important not to leave things uncited. For example, lines 41 to 48, to name just a few.
Response 2: I have added the following references:
- Chang, E.; Ghosh, N.; Yanni, D.; Lee, S.; Alexandru, D.; Mozaffar, T. A Review of Spasticity Treatments: Pharmacological and Interventional Approaches. Crit. Rev. Phys. Rehabil. Med. 2013, 25, 11–22, doi:10.1615/CritRevPhysRehabilMed.2013007945.
- Lee, K.W.A.; Chan, L.K.W.; Lee, A.W.K.; Lee, C.H.; Wan, J.; Yi, K.-H. Immunogenicity of Botulinum Toxin Type A in Different Clinical and Cosmetic Treatment, a Literature Review. Life 2024, 14, 1217, doi:10.3390/life14101217.
- Ploegmakers, D.J.M.; Van Duijnhoven, H.J.R.; Duraku, L.S.; Kurt, E.; Geurts, A.C.H.; De Jong, T. Efficacy of Selective Neurotomy for Focal Lower Limb Spasticity: A Systematic Review. J. Rehabil. Med. 2024, 56, jrm39947, doi:10.2340/jrm.v56.39947.
Comments 3: Introduction: Lines 63–64: paragraphs of only two lines are strongly discouraged.
Response 3: I combined it with the following paragraph. (Line 63-64)
Comments 4: Introduction: While it is true that you should report on what was unknown in previous meta-analyses in order to justify yours, this should be summarised in no more than six lines. There's no need to explain in depth what other authors did and what they did wrong – just summarise what they're missing and what you're going to address.
Response 4: To clarify the novelty of this meta-analysis compared to previous reviews, especially regarding methodology, scope, and clinical impact, I've increased the number of paragraphs. This conflicts with the opinions of other reviewers. We appreciate your understanding.
Comments 5: Methods: In addition to PRISMA, authors should follow and cite the Cochrane Collaboration Handbook.
Response 5: 2.1 Added reference [11] in the Study Design section. (Line 123)
Comments 6: Methods: The PRISMA Checklist should be included in the supplementary material.
Response 6: The PRISMA checklist has already been submitted to the academic editor and will be added as supplementary material.
Comments 7: Methods: “2.1. Study Design”: Lines 119-124: I don't understand why these lines are necessary, as they simply reiterate the objective and summarise the inclusion criteria, which are included later in the manuscript.
Response 7: I deleted it based on your opinion.
Comments 8: Methods: “The search language was limited to English.” With current translation tools (not including AI), there is little justification for limiting oneself to English, even though it is the dominant language in science.
Response 8: We acknowledge the reviewer’s concern regarding the restriction of the search language to English. While we recognize that modern translation tools can facilitate the inclusion of non-English literature, we opted for an English-only search primarily to ensure the accuracy of methodological assessment and data extraction, particularly for technical details related to vibration parameters and statistical methods. These aspects often require precise interpretation that may be compromised by automated translation. Moreover, the vast majority of high-quality randomized controlled trials on vibration therapy for post-stroke spasticity are published in English, as evidenced by preliminary scoping searches in multiple databases. Nevertheless, we agree that excluding non-English studies may introduce language bias, as noted in the Limitations section, and future research should seek to incorporate studies in other languages with the aid of professional translation and multilingual reviewers to enhance comprehensiveness and reduce potential bias. (Line 534-536)
Comments 9: Methods: “The inclusion of studies was independently assessed by two or more reviewers, and 141 disagreements were resolved through discussion or consultation with a third reviewer.”: two or more reviewers? Either there are two, or there are three, or there are… a specific number. Specify this, and who these two reviewers are. I also suggest changing the term "reviewer" to "author" (because I imagine the authors of the manuscript performed these procedures).
Response 9: I have revised it to two authors based on your comments. “The inclusion of studies was independently assessed by two authors (J.-W.Seo & J.-D.K)…” (Line 146-147)
Comments 10: Methods: Similarly, other sections should specify which authors performed each procedure (e.g. risk of bias assessment, data extraction) and their initials.
Response 10: The risk of bias assessment and data extraction were independently performed by three authors, including the corresponding author, and finalized through discussion. This procedure is specified in the “Author Contributions” section of the manuscript; therefore, we did not describe it separately in the main text. We appreciate your understanding. (Line 579-581)
Comments 11: Methods: The use of GRADE to assess the strength or quality of the evidence would enrich this systematic review and meta-analysis.
Response 11: We appreciate the reviewer’s suggestion to apply the GRADE framework. While we agree that GRADE can enhance the interpretability of systematic reviews, our study focused on quantitative synthesis and parameter-specific moderator analyses. Given the small number of studies in some subgroups and substantial heterogeneity, GRADE ratings would likely have been low across most domains, adding limited value beyond our Cochrane RoB 2 assessment and detailed heterogeneity analyses. We believe our current approach sufficiently informs the robustness of the findings, but we will consider GRADE in future research with a larger and more homogeneous evidence base.
Comments 12: Methods: “The random-effects model 220 was applied to integrate the overall effect size, and the effect size was calculated as Stand-221 ardized Mean Difference (SMD) and 95% confidence interval (CI) [14].”: What type of random effects were used? Dersimonian and Laird? Or which one?
Response 12: I have revised the sentence to read, “The random-effects model with the Restricted Maximum Likelihood (REML) method was applied to integrate the overall effect size, and the effect size was calculated as Standardized Mean Difference (SMD) and 95% confidence interval (CI) [14].” (Line 224–228)
Comments 13: Results: The PRISMA and AMSTAR-2 checklists require a list of excluded studies with the main reason for exclusion for each study. Include it in supplementary material.
Response 13: The PRISMA checklist has already been submitted to the editor and will be added as supplementary material.
Comments 14: Results: “Ultimately, 13 studies were included in this systematic review and meta-analysis.”: Authors should include citations here, as this is the first time they refer to all 13 included studies.
Response 14: I have included references based on your feedback. “Ultimately, 13 studies were included in this systematic review and meta-analysis [23-35].” (Line 265-266)
Comments 15: Results: Some of the studies included in the systematic review (references 1, 2, 3, 5, 6, 8) are used in the introduction. The studies included in the systematic review should never be included in the introduction. Occasionally, if imperative, it can be used to explain the need for your study. But, in general, it should not be used. I mean, for example, and this is just an example: “Spasticity is a frequent and debilitating sequela of stroke, characterized by a velocity-dependent increase in muscle tone due to hyperexcitability of the stretch reflex. This pathophysiological condition impairs voluntary movement, restricts joint mobility, and hinders functional recovery, thereby markedly diminishing patients’ quality of life.” You cannot use reference 1, which is a clinical trial that aimed to analyse the efficacy of an intervention. Appropriate references should be used, considering the objective of the referenced study. I don’t know if I’m making myself clear.
Response 14: References 1, 2, 3, 5, 6, and 8 have been replaced with the other references below.
1. Chan, K.-S.; Liu, C.-W.; Chen, T.-W.; Weng, M.-C.; Huang, M.-H.; Chen, C.-H. Effects of a Single Session of Whole Body Vibration on Ankle Plantarflexion Spasticity and Gait Performance in Patients with Chronic Stroke: A Randomized Controlled Trial. Clin. Rehabil. 2012, 26, 1087–1095, doi:10.1177/0269215512446314.
2. Brogårdh, C.; Flansbjer, U.-B.; Lexell, J. No Specific Effect of Whole-Body Vibration Training in Chronic Stroke: A Dou-ble-Blind Randomized Controlled Study. Arch. Phys. Med. Rehabil. 2012, 93, 253–258, doi:10.1016/j.apmr.2011.09.005.
3. Casale, R.; Damiani, C.; Maestri, R.; Fundarò, C.; Chimento, P.; Foti, C. Localized 100 Hz Vibration Improves Function and Reduces Upper Limb Spasticity: A Double-Blind Controlled Study.
5. Lee, J.-S.; Kim, C.-Y.; Kim, H.-D. Short-Term Effects of Whole-Body Vibration Combined with Task-Related Training on Upper Extremity Function, Spasticity, and Grip Strength in Subjects with Poststroke Hemiplegia: A Pilot Randomized Controlled Trial. Am. J. Phys. Med. Rehabil. 2016, 95, 608–617, doi:10.1097/phm.0000000000000454.
6. Seim, C.; Chen, B.; Han, C.; Vacek, D.; Wu, L.S.; Lansberg, M.; Okamura, A. Relief of Post-Stroke Spasticity with Acute Vi-brotactile Stimulation: Controlled Crossover Study of Muscle and Skin Stimulus Methods. Front. Hum. Neurosci. 2023, 17, doi:10.3389/fnhum.2023.1206027.
8. Marconi, B.; Filippi, G.M.; Koch, G.; Giacobbe, V.; Pecchioli, C.; Versace, V.; Camerota, F.; Saraceni, V.M.; Caltagirone, C. Long-Term Effects on Cortical Excitability and Motor Recovery Induced by Repeated Muscle Vibration in Chronic Stroke Pa-tients. Neurorehabil. Neural Repair 2011, 25, 48–60, doi:10.1177/1545968310376757.
Round 2
Reviewer 2 Report
Comments and Suggestions for Authors
Comments
The authors responded reasonably well to most of the comments, but there is still one thing they need to address.
- “Comments 13: Results: The PRISMA and AMSTAR-2 checklists require a list of excluded studies with the main reason for exclusion for each study. Include it in supplementary material.”: I mean, authors should include a list of excluded studies in the supplementary material, along with the main reason for their exclusion.
Author Response
Comments 1: The authors responded reasonably well to most of the comments, but there is still one thing they need to address.
“Comments 13: Results: The PRISMA and AMSTAR-2 checklists require a list of excluded studies with the main reason for exclusion for each study. Include it in supplementary material.”: I mean, authors should include a list of excluded studies in the supplementary material, along with the main reason for their exclusion.
Response 1: Thank you. We have attached the revised supplementary material including the list of excluded studies with reasons. We appreciate your valuable feedback.
